# Experimental Study on Geysers Induced by the Release of Trapped Air in Storage Tunnel Systems

**Xiaosheng Wang [1,2,\*], Shangtuo Qian [2] and Hongxun Chen [1]**

[1] Shanghai Key Laboratory of Mechanics in Energy Engineering, Shanghai Institute of Applied Mathematics and Mechanics, School of Mechanics and Engineering Science, Shanghai University, Shanghai 200072, China; chenhx@shu.edu.cn

[2] College of Agricultural Engineering, Hohai University, Nanjing 210098, China; qshttc@163.com

\* Correspondence: wxsh83@163.com

**Abstract:** A storage tunnel system is the critical infrastructure of urban drainage systems, in which the rapid filling of water and release of trapped air can lead to the "geyser" phenomenon. This may cause serious damage, threatening both system operation and personal safety. In this paper, a detailed experimental study was carried out based on synchronous recorded video images and digital image processing technology. According to experimental observations, gas-flow geysers and surge-type geysers were analyzed deeply. The former is caused by high-speed gas flow and is accompanied by a pressure drop; the latter is caused by surge pressure and is accompanied by a pressure increase. The free surface velocity of the gas-flow geyser is mainly affected by the external pressure, the air volume, the diameter of the shaft, and the height of the water column, and the geyser phenomenon cannot occur when the air column is mainly dominated by buoyancy. Based on dimensional analysis and data fitting, this paper presents the empirical formula for the free surface velocity and the interface net velocity and puts forward the critical occurrence conditions for the gas-flow geyser.

**Keywords:** storage tunnel; geyser; transient pressure; water–air interface; surge; Taylor bubble

## 1. Introduction

The storage tunnel system has been widely applied to urban drainage systems to effectively increase stormwater storage capacity and prevent waterlogging disasters. Rapid inflows into the drainage network may lead to water–air interactions that give rise to the "geyser" phenomenon [1–3] which has been reported several times in the past few decades. Geysers can cause water to be ejected out of the ground and can give rise to strong instantaneous pressure variation thus negatively impacting the operation of urban network systems.

Originally, Guo and Song [4] believed that surges caused by pressure oscillations were the main reason for geysers. Subsequently, Wright et al. [5] stated that pressure oscillations in the tunnel system alone were insufficient to give rise to geysers based on simple theoretical analysis. Lewis [6] also believed that the pressure oscillations induced by inertia could not fully explain the occurrence of the geyser phenomenon. Vasconcelos's experiment [7] indicated that the geyser phenomenon was closely related to the release of the trapped air pocket, and in the experiment, three types of geysers were observed: the air pocket released from the vertical shaft with initial water column, the water–air mixture jet, and the cross overflow between the air pocket and water. The first type was caused by the release of a large air pocket, and the last two types were caused by upstream surge. Huang et al. [8] divided the geysers into three types based on flow regime and pressure characteristics: short column jets, column-breaking jets, and spray-like jets. These geysers are driven by system pressure and are dominated by the water–air mixture flow.

Wright and Lewis [9] first elaborated the mechanism of the kind of geyser induced by air pocket release. When the moving trapped air in the tunnel reaches the vertical shaft position, the air pocket rises and drives the surrounding fluid to move upwards and break at the free surface. After the release of the air, the water levels in the vertical shaft reduce considerably and sustain inertial oscillations for a long time. Subsequently, Vasconcelos and Wright [10,11] carried out further research on this type of geyser, analyzing the influence of parameters, such as the water level of the shaft, air pocket pressure, and diameter of the shaft, and pointing out that the diameter is the key factor affecting the geyser strength.

Similarly, Leon et al. [12,13] described the geyser mechanism and divided the geyser process into six stages. The authors presented gas mass flow rate as a significant parameter that affects geyser strength and provided the empirical formula for the maximum geyser height based on dimensional analysis and experimental statistics data. Shao [14] analyzed the main driving factors of the geyser, including buoyancy, compressibility of air mass, inertia, and pressure, using two-dimensional numerical simulation and pointed out that the main factors affecting geyser strength are the volume of air pocket, the pressure in the main pipe, and the water level of the vent hole. Muller et al. [15,16] presented the relative relation between the water–air interface velocity and the free surface velocity using a large-scale experiment and computational fluid dynamics (CFD) simulation, and the results indicated that the velocities are approximately linearly proportional. Choi et al. [17] divided the geyser process into three stages, namely, the occurrence of pressure bore, the entrapment and migration of compressed air pocket, and the eruption of water–gas mixed flow, and then developed a schematic diagram to illustrate the relations between dominant variables associated with geysers using the modified Preissmann slot model [7,18] and the momentum equation.

Recently, Cong et al. [19] observed the release process of the air pocket by high-speed video camera technology. Based on the results, they provided another definition for a geyser in which the air pocket velocity is greater than the Taylor bubble velocity. They also identified the critical conditions for geyser occurrence based on statistical data. Chan et al. [20] conducted a numerical simulation and reached the same conclusion based on the volume of fluid (VOF) method and the standard k-$\varepsilon$ turbulence model.

In conclusion, previous scholars have conducted numerous studies on the geyser phenomenon based on experiments and CFDs. However, so far, these studies mainly focused on the phenomenal description and the analysis of the occurrence mechanism, and there is still a lack of analysis on the internal influencing factors of different types of geysers. In this paper, we discuss an experimental study on geysers caused by the release of trapped air, aimed at analyzing hydraulic characteristics of geysers including flow regime, instantaneous pressure, and interface characteristics. Then, we analyzed the mechanisms of different types of geysers and explored the free surface velocity, as these are crucial parameters for predicting geyser strength and geyser occurrence conditions.

## 2. Experiment Setup

### 2.1. Experimental Facility

In this paper, the method of capsule experiment was adopted which has been proved feasible by Lewis and Cong et al. [5,19]. The geyser experimental device is shown in Figure 1. The device consisted of a pressure water tank, tunnel (horizontal pipe), shaft (vertical pipe), and knife gate valves, where the size of the water tank was 2.00 m × 1.60 m × 1.20 m, the length of the upstream segment was 8.0 m, the length of the downstream section was 6.0 m, the tunnel diameter $D_0$ was a fixed value of 0.15 m, the height of shaft $H$ was 1.35 m, the diameter of shaft $D$ ranged from 0.05 m to 0.15 m, the volume of trapped air pocket could be controlled by adjusting the valves, the length of the trapped air pocket $L_0$ ranged from 0.60 m to 2.40 m, and the volume of the air pocket $V$ was $\pi D_0^2 L_0/4$. In order to facilitate observation, the tunnel and shaft were made of acrylic materials. In the experimental device, water was supplied from a submerged pump to the water tank, in which an adjustable overflow weir was set

to keep the water head stable. A total of four gate valves were arranged in the device. The valve at the upstream side was used to control external pressure, and the three valves at the downstream side were used to control different volumes of air pocket. The gate valves can be opened quickly by manual pulling. According to the statistics of multiple tests, the average opening time of a gate valve is around 0.25 s.

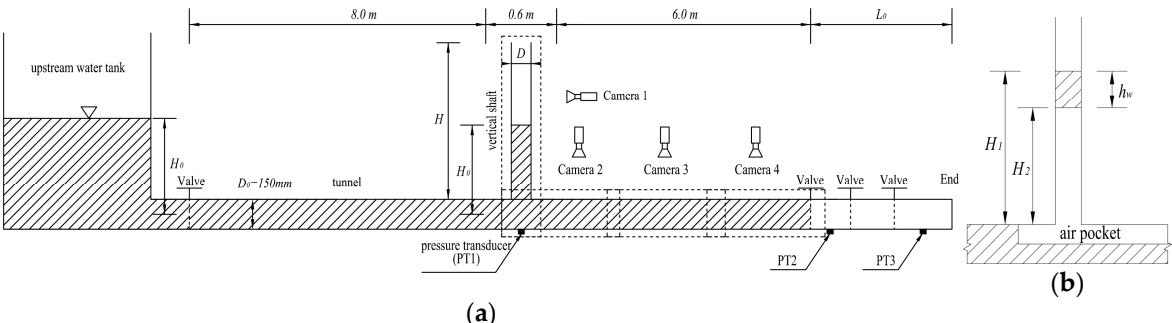

**Figure 1.** Schematic diagram of the storage tunnel system experimental device: (**a**) experimental device; (**b**) free surface and water–air interface.

## 2.2. Measurement Technique

Based on synchronous recorded video images and digital image processing technology, this paper measured the migration and release process of the trapped air pocket. The measurement system was composed of a computer, a data acquisition card, and four industrial cameras (2560 × 1024 pixels, 30 frames/second). Due to the radial and tangential distortion of the images, the initial images recorded by the camera were calibrated through the MATLAB Camera Calibration toolbox. A horizontal ruler and a vertical ruler were arranged in the test model to match the predetermined coordinates on the image. Then, the original image with distortion was corrected using the calibration parameters in the toolbox. Video images were converted according to the relation between image unit (pixel) and actual size (mm) so as to determine the position of the water–air interface and its migration speed. This value was approximately 1 mm per pixel in this experiment.

Transient pressure was collected by a dynamic signal acquisition system, and the transducer was a high-frequency dynamic pressure transducer (measuring range: $-3$ mH$_2$O to $+3$ mH$_2$O, measuring accuracy: 0.5% FS). In this experiment, the test frequency was 256 Hz, and three transducers were arranged in the tunnel, where the measurement point PT1 was used to test the pressure of the shaft bottom, and PT2 and PT3 were used to test the pressure of the air pocket in its initial position.

## 2.3. Experimental Conditions

The experimental device can operate in the following two modes:

- Mode A: The upstream valve is closed, there is no external pressure in the system, and the air pocket in the shaft can only be affected by buoyancy.
- Mode B: The upstream valve is open, there exists external pressure in the system, and the air pocket in the shaft is subjected to buoyancy and external pressure.

The operation parameters included diameter of shaft, length of air pocket, and water head of the tank, all of which can be adjusted as needed. In this experiment, 8 groups of experiments in mode A and 64 groups of experiments in mode B were carried out.

The operating conditions of the experiments are shown in Table 1.

**Table 1.** The operation conditions of the geyser experiment.

| Operation Mode | Diameter of Shaft, $D$ (m) | Length of Air Pocket, $L_0$ (m) | Water Head of Tank/Water Depth of Shaft, $H_0$ (m) |
|---|---|---|---|
| A | 0.05 | 0.60; 1.20; 1.80; 2.40 | 0.525; 0.675 |
| B | 0.05; 0.075; 0.10; 0.15 | 0.60; 1.20; 1.80; 2.40 | 0.225; 0.375; 0.525; 0.675 |

## 3. Results and Discussion

### 3.1. Flow Regime

Figures 2 and 3 show the typical flow regime of the tunnel and the geyser flow regime during the release of the trapped air pocket, respectively. After the control valve upstream was opened, the trapped air moved to the side of the shaft. The air pocket head was a smooth, curved water–air interface, as shown in Figure 2a, and the shape was consistent with the interface of the air intrusion water body described by Benjamin [21]. The propagation velocity of the air pocket head was approximately $0.52 \sqrt{gD_0}$. The tail of the air pocket was a free surface bore formed by reflected wave due to the water flow striking the end wall as shown in Figure 2b. The interface of the bore was highly aerated flow, and the tail has a long and narrow air chamber. The head shape and tail shape of the air pockets in mode A and mode B were consistent. According to Benjamin's [21] theoretical analysis, the velocity of air pocket propagation in stagnant water depends on the shape of the air pocket, and is independent of external pressure; therefore, the velocity of air pocket propagation under two modes has no significant discrepancy.

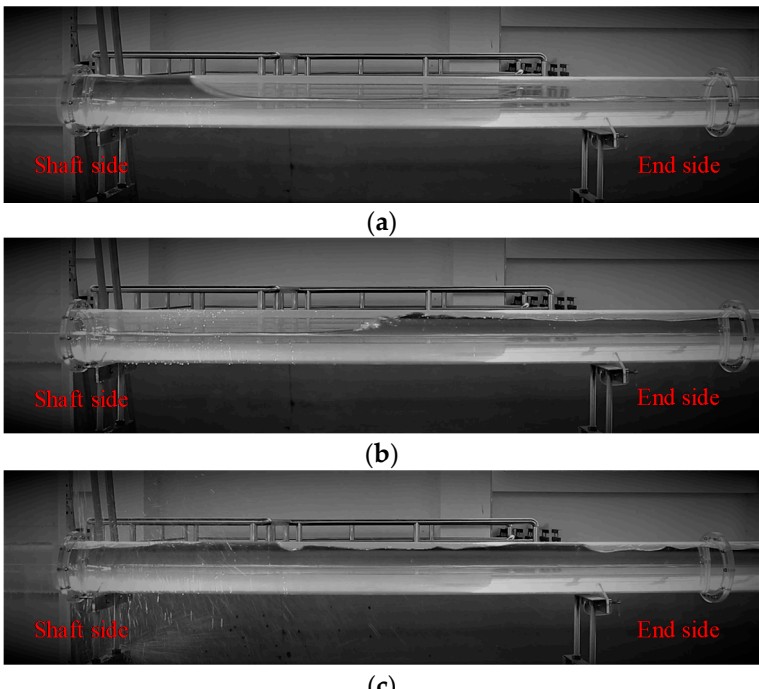

**Figure 2.** Typical flow regime in tunnel: (**a**) head of the air pocket; (**b**) tail of the air pocket; (**c**) unstable free surface waves.

When the air pocket head moves to the shaft location, under the effect of buoyancy, some of the air enters the shaft and forms an air column. As the amount of air entering the shaft gradually increases, the length of the water column in the shaft decreases accordingly, and the rise speed of the water–air interface gradually increases. The upper remnant water column is rapidly lifted and ejected out of the shaft in the role of high-speed gas flow. Under the effect of inertia, the ejection continues to rise and

spread to the surrounding air. The geyser evolution process is shown in Figure 3a–d. The geyser is mainly caused by gas-flow shock, so it is called a "gas-flow geyser".

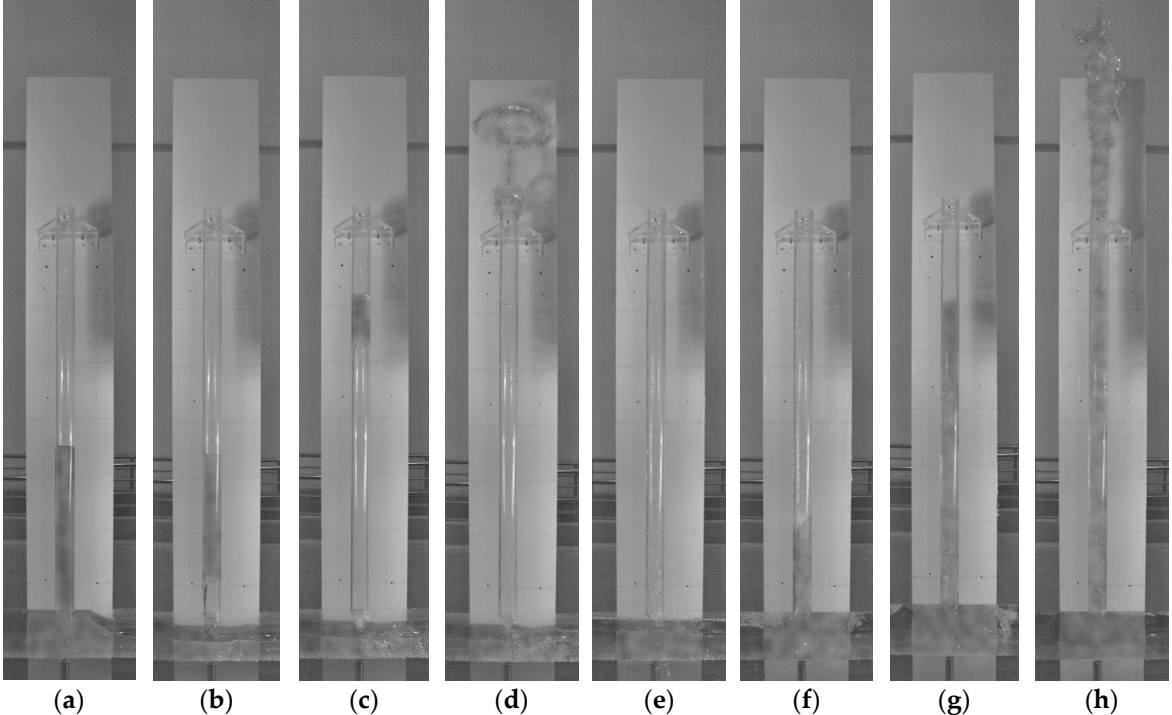

| (a) | (b) | (c) | (d) | (e) | (f) | (g) | (h) |

**Figure 3.** Geyser flow regime during the release of trapped air pockets. Operation condition: mode B, $D = 0.05$ m, $H_0 = 0.525$ m, $L_0 = 1.8$ m. (**a**) t = 11.60 s; (**b**) t = 12.16 s; (**c**) t = 13.08 s; (**d**) t = 13.44 s; (**e**) t = 14.56 s; (**f**) t = 14.80 s; (**g**) t = 15.00 s; (**h**) t = 15.52 s.

After the gas-flow geyser occurs, the air column is broken, and the pressure at the shaft position drops sharply. When the air pocket volume is large enough, the air pocket in the tunnel will be connected to the atmosphere, and the water flow in the tunnel will translate into open channel flow. When the air pocket volume is small, the air column in the shaft and the air pocket in the tunnel are not connected, but the local low-pressure area is still formed at the shaft position. Under the action of external pressure, water from a high-pressure water tank will rapidly fill the tunnel, quickly driving residual air out of the tunnel and significantly increasing the air discharge. When the water level in the tunnel is close to the pipe crown, the system pressure fluctuation intensifies the instability of the free surface and produces unstable wave flow and slug flow in the tunnel (as shown in Figure 2c). Under the action of inertia, the water flow continues to fill into the tunnel and strike the water–air mixture, which produces a water hammer effect and causes the system pressure to surge. Subjected to instantaneous high pressure, the water–air mixture pours into the shaft and rises rapidly. When the extreme pressure is large enough, the water–air mixture will jet out of the shaft and form a geyser. The geyser evolution process is shown in Figure 3e–h. The geyser is caused by the instantaneous pressure surge of the system, so it is called the "surge-type geyser".

Under operation mode A, there is no external pressure, so an air column can be formed in the shaft, but the interface velocity is slow, and the air column immediately breaks down in a short time, so the maximum height of the free surface level is equivalent to the initial water level. Therefore, a geyser is unable to form. After the air column is broken, the water in the tunnel tends to rapidly become stable without pressure surge due to the absence of external head and no geyser occurrence.

### 3.2. Transient Pressure

Figure 4 shows the variation of system transient pressure during the release of trapped air pockets under operation modes A and B, in which, $P/\rho_w g$ represents the pressure head.

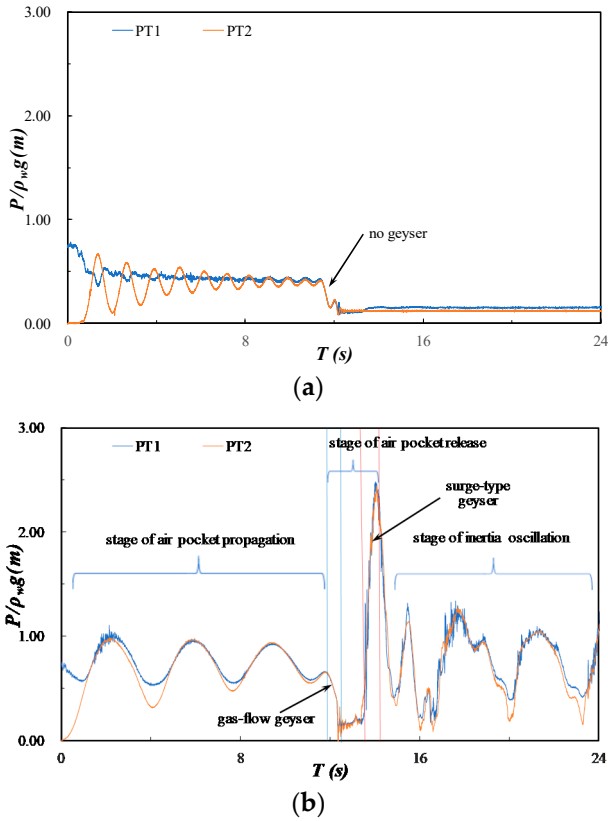

**Figure 4.** Variation of transient pressure during the release of a trapped air pocket. (**a**) Operation condition: mode A, $D$ = 0.05 m, $H_0$ = 0.675 m, $L_0$ = 2.4 m; (**b**) operation condition: mode B, $D$ = 0.05 m, $H_0$ = 0.675 m, $L_0$ = 2.4 m.

According to the transient pressure variation characteristic, the air pocket release process can be divided into three stages: the stage of air pocket propagation in the tunnel, the stage of air pocket release, and the stage of inertia oscillation, among which the geysers occur in the stage of air pocket release.

At the stage of air pocket propagation in the tunnel, affected by air pocket compressibility, the system pressure presents a law of periodic oscillation. Under mode A, the pressure of PT1 is significantly reduced due to the reduction of the water level in the shaft. As the water level of the shaft gradually stabilizes, the period of pressure oscillation is short, and the amplitude decays quickly. The pressure of PT2 increases rapidly from zero and maintains wide oscillation due to the compression and expansion effect. As the air pocket propagates downstream, the pressure of two measuring points gradually tends to be consistent. On the contrary, in mode B, under the excitation of the external pressure head, the compression and expansion effect of the air pocket is significantly enhanced, the pressure oscillation period of each measuring point increases, and the amplitude attenuation rate becomes rather slow. The pressure of the two measuring points was consistent.

At the stage of air pocket release, with the length of the air column increasing gradually, the system pressure drops rapidly. The greater the gas flow velocity, the faster the pressure drop. When the air column is broken, the system pressure drops to a minimum. At this moment, the water flow in the tunnel transforms into open channel flow, and the system pressure maintains a high-frequency oscillation at a low pressure.

During the process of the air column rising and breaking, the tendency of pressure variation in mode A is the same as that in mode B, but no geyser is formed, which indicates that the pressure drop is mainly caused by the rise of the air column and the decrease of the water column length, and the external pressure plays an important role in the formation of the gas-flow geyser.

When the water–air mixture is near the tunnel crown, it will be squeezed, and the system pressure increases rapidly. The maximum pressure is up to 3–5 times the external pressure. In mode A, no pressure surge can be generated due to the absence of external pressure.

After a geyser occurs, the instantaneous high pressure in the system is released, and the system pressure continues to oscillate periodically under the action of external pressure and inertia, but the oscillation amplitude is significantly lower than the maximum surge pressure. In this stage, the air pocket is rarely released. When the residual air content in the system is still relatively high, under the action of pressure oscillation and water–air interaction, multiple geysers may be produced.

### 3.3. Free Surface and Interface

Firstly, we defined the following parameters: $H_1$ is the level of the free surface, $H_2$ is the level of the water–air interface head, $h_0$ is the length of the water column at the initial moment when the air pocket head reaches the shaft position, and $H_{net}$ is the net motion of the water–air interface relative to $h_0$ which can be understood as the reduced length of the water column due to the rise of the air column. The relationships among these four variables can be defined as follows: $H_{net} = h_0 - (H_1 - H_2)$. Correspondingly, we establish that $v_1$ and $v_2$ are the velocity of the free surface and water–air interface, respectively, and $v_{net}$ is the net velocity of the water–air interface. Consequently, $v_1 = dH_1/d_t$, $v_2 = dH_2/d_t$, and $v_{net} = v_2 - v_1$.

Figure 5 shows the level variation of the free surface and water–air interface in the process of gas-flow geyser. In operation mode A, there is no external pressure, the air column rises only depending on its own buoyancy, the free surface and interface are still affected by the system pressure oscillation, the free surface has a level oscillation phenomenon, and the lift height of the free surface level is small. The value of $H_{net}$ increases roughly linearly along with time, and the corresponding velocity is $v_{net} = 0.25$ m/s which is approximated with the Taylor bubble velocity provided originally by Davies and Taylor [22] as $v_{taylor} = 0.35\sqrt{gD} = 0.245$ m/s. For a Taylor bubble, the free surface level remains constant when the rising bubble is subject only to buoyancy action. Likewise, in geyser flow, when the air pocket is dominated by buoyancy, the variation of the free surface level is mainly caused by air compression and expansion. Thus, the maximum level of the free surface is equivalent to the maximum height of level oscillation.

As shown in Figure 5b, in operation mode B, the free surface level and interface level both exhibit accelerating rise. The $H_{net}$ value increases roughly linearly along with time, but the $v_{net}$ value is significantly greater than the Taylor bubble velocity. This shows that external pressure has an obvious role in causing high-speed gas flow; this may be the major reason for the development of gas-flow geysers.

Figure 6 shows the variation of the free surface level and pressure head in the process of surge-type geyser, where $H_3$ is the free surface level relative to the PT1, and $H_p$ is the pressure head of the measuring point PT1. In the process of a surge-type geyser, firstly, the pressure rises rapidly and reaches the highest value within a short period of time, and the rise of the free surface level significantly delays the increase of pressure which also proves that the surge-type geyser is caused by the instantaneous high pressure of the system. Because a large amount of air is carried into the water during the surge process, the water–air mixture in the shaft is subject to pressure and buoyancy. The rising height of the free surface level is significantly higher than the extreme value of instantaneous pressure, and the difference value depends on the air content of the mixture in the shaft.

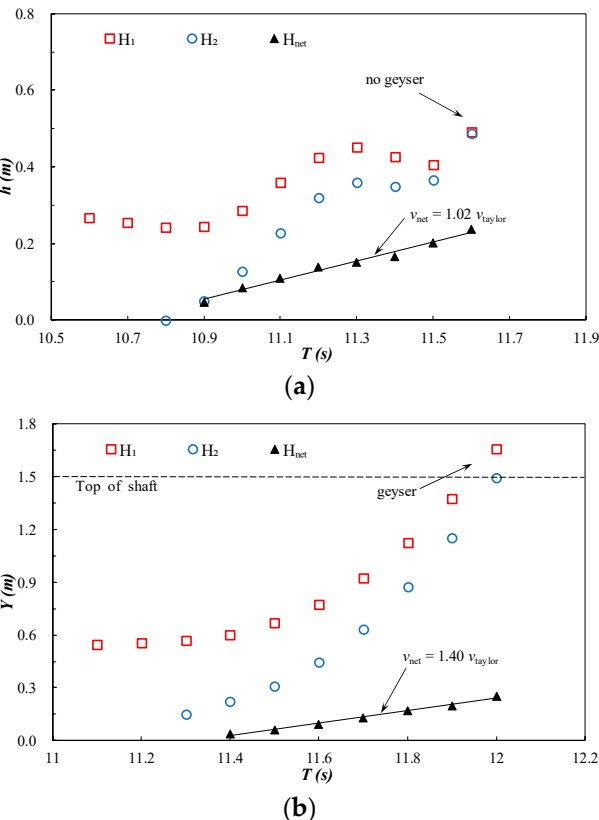

**Figure 5.** Level variation of free surface and interface in the process of a gas-flow geyser. (**a**) Operation condition: mode A, $D = 0.05$ m, $H_0 = 0.675$ m, $L_0 = 1.2$ m; (**b**) Operation condition: mode B, $D = 0.05$ m, $H_0 = 0.675$ m, $L_0 = 2.4$ m.

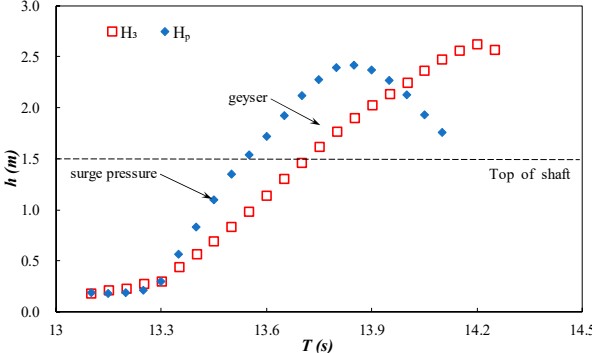

**Figure 6.** Level variation of free surface in the process of a surge-type geyser. Operation condition: mode B, $D = 0.05$ m, $H_0 = 0.675$ m, $L_0 = 2.4$ m.

## 4. Data Analysis and Discussion

A surge-type geyser is caused mainly by instantaneous surge pressure. Referring to the working principle of the water hammer pump, when there is instantaneous extreme pressure in the system, the pressure can send water to a higher position. If considering the water–air mixture, the lifting height will increase due to the density difference. Surge-type geyser height mainly depends on the value of instantaneous extreme pressure, and the instantaneous pressure normally can be analyzed using the classical water hammer theory. Previous studies [5,8] have carried out some detailed analysis for this kind of geyser, so it is not discussed in this paper.

In the process of gas-flow geyser development, it is difficult to analyze the instantaneous velocity feature. However, through analyzing the average free surface velocity, $\bar{v}_1$, and average interface net

velocity, $\bar{v}_{net}$, from a certain height of the water column, $h_w$, to 0 ($h_w = 0$ means the air column is broken), we can effectively estimate the intensity of gas-flow geyser and its occurrence conditions.

Gas-flow geyser development is mainly dominated by buoyancy, external pressure, and air compression and expansion. The influencing factors are mainly the physical constants and other parameters, including the water kinematic viscosity ($v_w$), water density ($\rho_w$), surface tension ($\sigma$), acceleration of gravity ($g$), external pressure ($H_0$), air volume ($V$), diameter of shaft ($D$), diameter of tunnel ($D_0$), height of water column ($h_w$), etc. The average free surface velocity can be expressed as:

$$\bar{v}_1 = \varphi(\rho_w, v_w, \sigma, g, H_0, V, D, D_0, h_w) \tag{1}$$

There are 10 variables in Equation (1) and three basic dimensions (mass, length, and time). Thus, seven dimensionless terms can be obtained, making length variables dimensionless, and the average free surface velocity can be expressed as:

$$f\left(\frac{\bar{v}_1}{\sqrt{gD}}, \frac{(\sqrt{gD})D}{v_w}, \frac{\rho_w(\sqrt{gD})^2 D}{\sigma}, \frac{H_0}{D_0}, \frac{V}{\pi D_0{}^3/4}, \frac{D}{D_0}, \frac{h_w}{D_0}\right) = 0 \tag{2}$$

The first dimensionless term on the left side of Equation (2) is the Froude number (*Fr*), the second and third dimensionless terms are equivalent to the Reynolds number (*Re*) and the Weber number (*We*), the term $\sqrt{gD}$ can be thought as a characteristic velocity. As indicated by Pfister and Chanson [23], the influence of kinematic viscosity and surface tension can be ignored when $Re > 3 \times 10^5$ and $We > 1.2 \times 10^4$. In this paper, the free surface velocity at the moment of air column breaking under typical condition (e.g., $D = 0.05$ m, $H_0 = 0.675$ m, $L_0 = 2.4$ m) is approximately 6.8 m/s; this can represent the characteristic velocity of a geyser, $v_w \approx 10^{-6}$ m$^2$/s and $\sigma \approx 0.072$ N/m, the corresponding *Re* and *We* are $3.4 \times 10^5$ and $3.2 \times 10^4$, which are larger than the above limiting values. Thus, the second and third dimensionless terms can be neglected, and the relevant dimensionless terms for the Froude number *Fr* of the free surface can be reduced to:

$$Fr = \frac{\bar{v}_1}{\sqrt{gD}} = \alpha \left(\frac{H_0}{D_0}\right)^{\alpha 1} \left(\frac{V}{\pi D_0{}^3/4}\right)^{\alpha 2} \left(\frac{D}{D_0}\right)^{\alpha 3} \left(\frac{h_w}{D_0}\right)^{\alpha 4} \tag{3}$$

where $\alpha$, $\alpha 1$, $\alpha 2$, $\alpha 3$, and $\alpha 4$ are empirical constants. In this paper, the $\bar{v}_1$ value is statistically analyzed from $h_w = 0.15$ m to $h_w = 0$ with no geyser conditions. Through regression analysis, one excellent fitting curve with high consistency was obtained (as shown in Figure 7). The corresponding index values were $\alpha = 0.008$, $\alpha 1 = 1.5$, $\alpha 2 = 0.5$, and $\alpha 3 = -2.5$. The *Fr* of free surface is expressed as:

$$Fr = \frac{\bar{v}_1}{\sqrt{gD}} = 0.008 \frac{H_0^{*1.5} V^{*0.5}}{D^{*2.5}} \tag{4}$$

where $H_0^* = H_0/D_0$, $V^* = V/\pi D_0{}^3/4$, $D^* = D/D_0$. When considering the influence of water column height, under the geyser condition, the air column is not broken in the shaft, the $\bar{v}_1$ value is obtained by extrapolation according to the velocity variation trend. Vast data on free surface velocity, including geyser cases and no geyser cases, were analyzed, and the fitting curve was obtained as shown in Figure 8. The corresponding equation is expressed as:

$$Fr = \frac{\bar{v}_1}{\sqrt{gD}} = 0.009 \frac{H_0^{*1.5} V^{*0.5}}{D^{*2.5} h_w^{*0.5}} \tag{5}$$

where $h_w^* = h_w/D_0$. However, when the height of the water column is greater than the external pressure head, the free surface cannot be moved upward, so the formula is inappropriate. When the

height of the water column is relatively small, the disturbance of the water column is enhanced. This may increase the deviation of the free surface velocity.

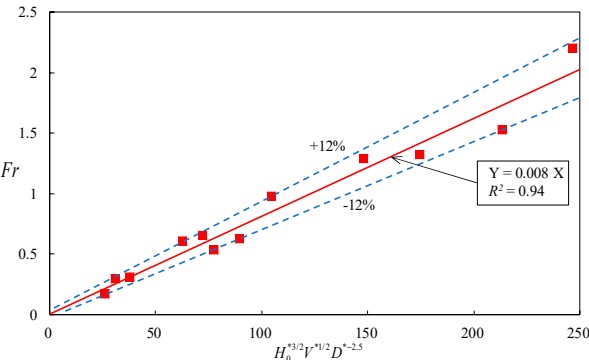

**Figure 7.** Fitting curve for the Froude number of average free surface velocity.

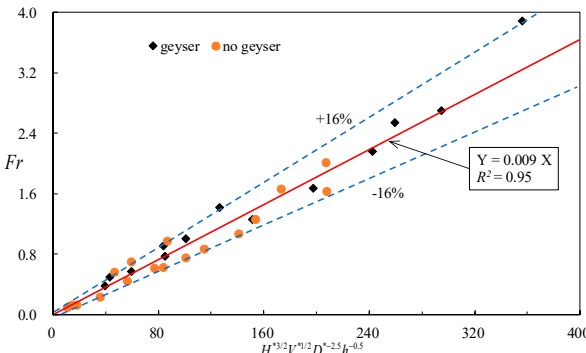

**Figure 8.** Expanded fitting curve for the Froude number of average free surface velocity.

For the average interface net velocity, $\overline{v}_{net}$, which is mainly dominated by the flow characteristics of the liquid film, its value floats at a fixed constant during the gas-flow geyser. The influencing factors are relatively complex, the free surface velocity ($v_1$) and the value of $D$ have the most significant influence. Figure 9 shows the relation between the Froude number of average interface net velocity, $Fr_{net}$, and the Froude number of free surface velocity, $Fr_1$. The corresponding equation is defined by data fitting as:

$$Fr_{net} = \frac{\overline{v}_{net}}{\sqrt{gD}} = 0.025 Fr_1 + 0.39 \tag{6}$$

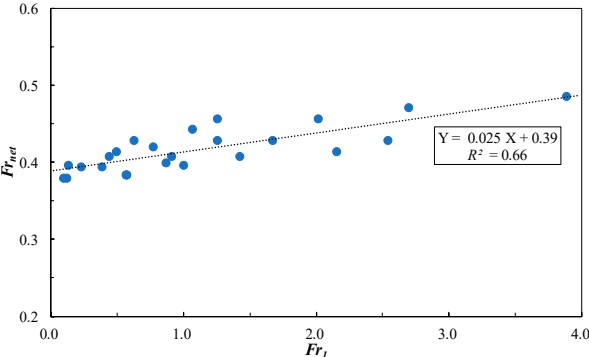

**Figure 9.** Comparison between $Fr_{net}$ and $Fr_1$.

According to the expressions of $\bar{v}_1$ and $\bar{v}_{net}$, when the vertical shaft diameter is large, then the free surface velocity will be small, and the net velocity of the interface will be large, and it will be more difficult to form a gas-flow geyser. Meanwhile, by comparing the relative relation of the $\bar{v}_1$ value, $\bar{v}_{net}$ value, the height of shaft $H$, and the initial height of the water column $h_0$, the occurrence conditions of gas-flow geyser can be obtained as follows:

$$\frac{\bar{v}_1}{\bar{v}_{net}} = \frac{Fr_1}{Fr_{net}} > \frac{H}{h_0} - 1 \tag{7}$$

Figure 10 shows the relation between $\frac{Fr_1}{Fr_{net}}$ and $\frac{H}{h_0} - 1$ based on fitting Formulas (5) and (6). The results are in good agreement with the experimental results. The upper part of the figure is the geyser region, the bottom part is the no-geyser region, the middle part between dotted lines is the transition region, the calculation value in the transition region may have a certain degree of error because of the unstable characteristic of the geyser, and the deviation was between 10% and 20%.

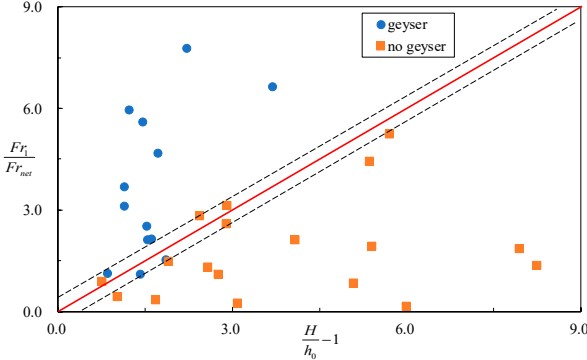

**Figure 10.** Diagram of occurrence conditions of gas-flow geysers.

However, in this paper, the tunnel diameter of the experiment device was fixed at 0.15 m, the effect of tunnel diameter on the geyser was not determined, and the conclusions presented in this paper still need to be further verified. In addition, the $v_{net}$ value was extremely unstable; the shape of the air column had a great influence on the $v_{net}$ value. When the air column head has a deflective or unsteady interface, the $v_{net}$ value increases significantly, so the measured data becomes more random and less regular. It is necessary to conduct more investigations on this topic by experimental or CFDs methods in the future.

## 5. Conclusions

In this paper, a detailed experimental study was carried out on geysers induced by the release of trapped air pockets. The characteristics of flow regime, transient pressure, free surface, and water–air interface were analyzed in detail.

According to the different features of geysers and their formation mechanisms, geysers can be divided into two types: gas-flow and surge-type. The former is caused by high-speed gas flow impinging on the upper water column due to the release of an air pocket which manifests as water–air interface flow. The latter is caused by surge pressure due to the water hammer effect which manifests as a jet of water–air mixture. For the gas-flow geyser, the head shape of the water–air interface is similar to a Taylor bubble, and the system pressure gradually drops in the geyser process until the air column is broken. On the contrary, in the process of a surge-type geyser, the system pressure gradually increases.

Among the two types of geysers, the free surface motion of the gas-flow geyser is more complicated. When the air column is mainly dominated by buoyancy, the $v_{net}$ value is close to the Taylor bubble velocity, and no geyser is formed. When there is external pressure in the system, the $v_{net}$ value is

significantly greater than the Taylor bubble velocity which may be the main reason for gas-flow geyser formation. In the process of gas-flow geyser development, the free surface velocity is mainly affected by the external pressure, the air volume, the diameter of the shaft, and the height of the water column. The influencing factors of interface net velocity mainly included the free surface velocity and the diameter of the vertical shaft.

**Author Contributions:** Conceptualization, X.W.; Data curation, X.W. and S.Q.; Formal analysis, X.W. and S.Q.; Funding acquisition, X.W.; Investigation, X.W.; Methodology, X.W.; Project administration, X.W.; Resources, X.W. and H.C.; Validation, X.W., H.C. and S.Q.; Visualization, X.W.; Writing—original draft, X.W.; Writing—review and editing, X.W. and H.C.

**Funding:** This research was supported by the Fundamental Research Funds for the Central Universities (No. 2018B45014).

**Acknowledgments:** We greatly appreciate the careful reviews and thoughtful suggestions by reviewers.

**Conflicts of Interest:** The authors declare no conflict of interest.

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
