# Peer review of "Experimental Study on Geysers Induced by the Release of Trapped Air in Storage Tunnel Systems"

_applsci, doi:10.3390/app9245326_

Round 1
Reviewer 1 Report
Photos on Figure 2 could be bigger because the water movement is not well visible.
For the rest I like the paper.
Author Response
Comments:Photos on Figure 2 could be bigger because the water movement is not well visible.
Response :The photos of Figure 2 has been enlarged in the paper.
Reviewer 2 Report
An experimental investigation on the geyser flow features when a geyser is originated by the release of trapped air pockets is described in the manuscript. Physical model observations are described, and the experimental results are elaborated to obtain some interesting criteria to estimate air water two phase flow velocity.
My general comments are here listed:
Abstract is too lengthy (about 220 words instead of the maximum of 200 as suggested by the Journal’s prescriptions). Moreover, it may be structured better in order to highlight methods and main results. Methods are almost absent in the abstract, whereas the description of the observed geyser types is too detailed. The latter can be shortened, leaving space to some additional details about the experimental method; Introduction is well organized but, in my opinion, it can be further improved. I would have added a very short outlook on the main results just at the end of this section. May the Authors briefly highlight the main outcomes derived from the present experimental investigation? As detailed in the specific comments, given your experimental data I am not sure that the dimensional analysis can neglect the effects of the surface tension and viscosity. I ask Authors, therefore, to verify carefully the limiting criteria suggested by Pfister and Chanson (2012) for physical modelling of air water two-phase flows and to provide more details in the main text; Subsection 3.4 should be transformed in a new section specifically dealing with the Discussion. Moreover, few lines at the end of the data analysis and discussion should be devoted to the future research directions. In the end, I think that organizing the description of the main outcomes by numbering the retained issues (as done in section 4.) is not “stylish” for research papers. I would also try to shorten Conclusions a little bit. For instance, the issue described at point (4) can be drastically summarized by only stating which main influencing factors in the geyser flow formation have to be considered.
The text contains some typos. Some of them are pointed out in the Specific comments.
English is not entirely satisfactory. The English grammar and, overall, the writing form need to be improved. Some sentences are written in a very poor English language level, indeed. Authors are thus invited to work further to enhance the English in the text.
Specific comments
In the text, references should be placed in square brackets [ ] without reporting Authors’ names; Line 62: “relationship” should be used to intend people connections, and not to refer to maths equations. Please, use relation in place of relationship. It is more rigorous; Lines 85-86: do not denominate parts of the tunnel as “section”. This is commonly used to indicate pipe and conduit dimensions and shapes transversally. As an alternative, use “segment” or “portion”; Figure 1: it is useful for readers interested in reproducing the experimental tests to report the position of the four cameras in the experimental facility sketch. May the Authors insert the cameras positions in Figure 1? Line 128: the control valve mentioned by Authors should be that one placed upstream; Figure 2: May the Authors report in the figures, or in the caption, the position of the end wall, on one extremity, and of the shaft, on the other one? Line 135: Did Authors mean that the air pocket propagation velocities in mode A and B were almost equal? Please make this issue clearer; Lines 138-139: the sentence “..the rise speed of the water-air interface gradually accelerates” is written in a poor English. A velocity value usually increases/augments, but it does not accelerate. Please, modify it; Line 157: What does “extrude with” mean? Line 169: Authors should specify that transient pressure is considered by using P/(ρwg). You should also specify the meaning of these variables; Lines 175-178: the variation of the pressure transient, as measured by the PT2, is not commented despite it is shown in Figure 4a. May you provide some comments about PT2 observations and about the differences between PT1 and PT2 measurements? Line 176: please, substitute “with” with “due to” in “…significantly reduced with the reduction…”. Moreover, the statement “the compression and expansion effect on the air pocket decreases” is unclear. Please, clarify this issue; Figure 4: Why did Authors not represent the air pocket release stage in Figure 4b, as instead done for propagation and inertial oscillation stages? Moreover, it would be convenient to adopt the same extreme values of P/(ρwg) on the y-axes of Figs. 4a and 4b. This would make the comparison between pressure transients of modes A and B easier. In the end, the caption of Fig. 4b reports two different values of Ho. Why? If one of them is referred to the water head in the shaft, please use different symbols to represent tank and shaft heads; Lines 203-208: it would more rigorous to introduce these parameters in section 2. Moreover, the reader would be helped by providing a sketch showing the free-surface and interface motion parameters. Authors might add a focus of Figure 1 (as Fig. 1b, for instance) where these parameters are graphically indicated; Lines 216-217: Authors observed that vnet = 0.25 m/s resulted to be approximately equal to vtaylor = 0.35 m/s. However, the trendline equation shown in Fig. 5a (vnet = 1.02∙vtaylor) does not match this statement. Why? Line 229: What did Authors stand for “mean specific value of vnet and vtaylor”? Line 233: “net” and “taylor” in vnet and vtaylor should be typeset as subscripts; It is definitely not clear which difference subsists between Fig. 5b and Fig. 6. They should be both referred to operational mode B, one for Lo = 2.4 m and the other for Lo = 1.8 m. They both show the time variation of H1, H2 and Hnet. If this is correct, then why does Figure 6, and not Fig. 5b, represent the unsteady feature of the interface velocity? Figure 5-6-7: please, specify the meaning of the dotted line denominated “top” in these plots; Subsection 3.4 (data analysis and discussion) should be converted in a new section 4, as suggested in the journal prescriptions (please see the Journal template); Line 268: substitute “terms” with “term”; Line 271: If I do not make a mistake (please, check the validity range) Pfister and Chanson (2012) suggested to model air-water two-phase flows by neglecting surface tension and viscosity effects for R > 2÷3∙105 and W > 1.2∙104. These criteria are also valid for large Froude numbers (F > 5), and they should be more conservative for F < 5, as in your case. Do your experimental data respect these limits? Please, motivate your reply and insert a brief discussion in the paper; Lines 300-302: here, the English is very poor and, consequently, the statement looks quite hard to be fully comprehended. Please, re-write this sentence; Figure 11: Authors did not provide the equations of the boundary lines of the transition region. Why? Lines 310-316: I would move this consideration in the Introduction, or in the first part of the Discussion section.

Author Response
Please see the attachment, thanks.
